# Common knowledge processing patterns in networks of different systems

Aviv Segev *, Sukhwan Jung

Department of Computer Science, University of South Alabama, Mobile, AL, United States of America

* segev@southalabama.edu

## Abstract

Knowledge processing has patterns which can be found in biological neuron activity and artificial neural networks. The work explores whether an underlying structure exists for knowledge which crosses domains. The results show common data processing patterns in biological systems and human-made knowledge-based systems, present examples of human-generated knowledge processing systems, such as artificial neural networks and research topic knowledge networks, and explore change of system patterns over time. The work analyzes nature-based systems, which are animal connectomes, and observes neuron circuitry of knowledge processing based on complexity of the knowledge processing system. The variety of domains and similarity in processing mechanisms raise the question: if it is common in natural and artificial systems to see this pattern-based knowledge processing, how unique is knowledge processing in humans.

## Introduction: Knowledge and brain

Thinking can be defined as manipulating information, reasoning, and making decisions. Thought is based on building patterns of knowledge in a system. Can there be an underlying structure that repeats across systems, when these systems process information for different purposes and seem to have different infrastructures?

The theory of knowledge has evolved over the last 2,500 years, starting from the idea that "knowledge is perception" [1] and that these perceptions are processed in the brain [2]. The neuron was identified as the smallest knowledge processing mechanism in the brain [3].

Today the brain is seen as a network of nodes and relations [4]. Each section of the brain forms a smaller network with specific functionality, and the different sections are connected together. Some sections of the brain have been identified to perform specific tasks. The vibrant research area of artificial neural networks was inspired by research on biological neural networks [5–7].

What if artificial neural network knowledge processing patterns are compared with knowledge processing patterns of a variety of animals that show different complexity of knowledge processing patterns, or in other words different brain complexity? Would there be any common characteristics?

There are a variety of methods for analyzing activity of multiple neurons. Some of these methods are based on three-dimensional imaging scanning of tissue [8–10]. Others include

**Data Availability Statement:** Combinations of existing and generated datasets in multidisciplinary fields were used in the experiment. The authors decided to keep the datasets separate instead of having them in a single repository based on the various natures of the datasets being used, and

therefore the datasets can be found in multiple locations including their original sources. All the datasets used in the experiment are publicly available. Three existing multilabel classification datasets were used for reviewing the Patterns of Decision Making in Artificial Neural Networks, where a series of images are classified into a set of labels. They are all publicly available. • MNIST (http://yann.lecun.com/exdb/mnist/): Handwritten digits • Fasion_MNIST (https://github.com/zalandoresearch/fashion-mnist): Fashion items • KMNIST (http://codh.rois.ac.jp/kmnist/index.html.en): Hiragana (Japanese characters) Analysis of the Information Processing Mechanism on the Academic Topic Networks is done on eight journal-specific datasets curated from the Microsoft Academic Graph bibliography database. The curated dataset is made public in the following location. • Domain-specific Topic Co-occurrence Networks from Microsoft Academic Graph (https://zenodo.org/record/6547761). Similar patterns in randomly generated graphs were reviewed from fixed graphs generated for the experiment. The generated graphs were recorded online for public access. • Four random graph generation algorithms with example graph binary files (https://zenodo.org/record/6547767). The knowledge processing patterns in animal neuron-synapse structures were analyzed based on two existing neural map connectomes, which were made public by respective research publications. • Roundworm connectome (http://www.wormatlas.org/handbook/nshandbook.htm/nswiring.htm): Outdated, from Paper "Wiring optimization can relate neuronal structure and function". • Fruit fly hemisphere connectome (https://neuprint.janelia.org/): From Paper "A connectome of the adult drosophila central brain". Lastly, the minimal data set is uploaded to an online data repository for public access. • Common Knowledge Processing Patterns in Networks of Different Systems Minimal Data Set (https://zenodo.org/record/8122402). Corresponding author email: segev@southalabama.edu.

**Funding:** The authors received no specific funding for this work.

**Competing interests:** The authors have declared that no competing interests exist.

mapping complete neural population in specific organs [11, 12]. Many techniques analyze animals by the number of neurons and try to place certain behaviors with their neural network [13, 14].

The neural network connectivity and the network behavior analysis can consider the interaction signal activity of the neurons. The resolution of time can be measured in one-thousandth of a second to longer time periods which could be up to weeks or even months. Similarly the area analyzed can be from one-thousandth of a centimeter to a few centimeters [15, 16].

Emission of electromagnetic radiation in the neural network is used to analyze the process of the neural activity and the connected activity. The emission is used to process the activity of different sizes of neural networks and their associated common simultaneous activity. Neural network spiking is processed as a single neuron exciting multiple neurons that are not necessarily connected [17]. Background activity can also be analyzed as a cause for neural network spiking activity [18, 19].

The work proposes the idea that there are underlying characteristics such as the connectivity structure between the nodes of the system processing the information. These underlying patterns of characteristics appear to be consistent between the different systems as they evolve and successfully process more sophisticated information. Biology has DNA as an underlying basic element characteristic, Chemistry has the Periodic Table, and Physics has a set of equations. This work analyzes whether knowledge has an underlying structure, "DNA for Knowledge".

The main contributions are:

- A new knowledge modeling approach is presented based on evolving graphical characteristics.

- The knowledge modeling approach is analyzed on different network processing systems: artificial neural networks, academic topic networks, graph generation algorithms, and animal neuron-synapse structures.

- From the different graphical evolving characteristics some similarities appear across the different network processing systems.

The next section provides background information on related work on knowledge processing in neuronless, non-biological systems and is followed by the methods section describing the knowledge processing comparison analysis and the experiments on different network processing systems. Then the results section identifies the characteristics that appear in the different processing systems. The discussion section presents the overall commonality of the knowledge model across the processing systems. Finally, the conclusion summarizes the results and presents future work.

## Related work

### Machine learning and nature-inspired algorithms

Machine learning has been extensively used in areas such as content-based medical image retrieval [20], topic classification of online news articles [21], and prediction of solar energy generation [22]. Multiple approaches have been used to simplify the processing of the neural network. A swarm intelligence-based classification algorithm was proposed for reducing dimensionality (feature selection) in datasets [23]. Other approaches use heterogeneous graph embedding to learn the low-dimensional representations of nodes, such as link prediction, node classification, and community detection [24]. Furthermore, community detection was performed by mapping nodes into communities based on a random walk in the network [25].

Nature-inspired machine learning algorithms appear in characterization of abnormalities in breast cancer images involving training a Convolutional Neural Network using genetic algorithm (GA), whale optimization algorithm (WOA), and multiverse optimizer (MVO), satin bower optimization (SBO), and life choice-based optimization (LCBO) algorithms to optimize weights and bias of the model [26]. In addition, COVID-19 cases prediction was performed using a hybrid machine learning and beetle antennae search approach [27].

## Neuronless knowledge processing

Decision making can be viewed in a collection of trees, a forest, and previous work compared the input and output of each tree to the activity of a neuron. Biologically engineered knowledge processing capabilities can be viewed as networks of neurons of plants. Existing literature on information sharing, reasoning, and decision making in a collection of a large number of plants, or forests is reviewed. The comparison showed similarity between the mechanism based on neurons and the mechanism based on trees.

There has been research on similar neuron structure and neuron transmitter structure in plants [28, 29]. Although neurons are viewed as the smallest unit which processes and transmits information, there have been no findings leading to support that brain-like or neuron-like structure appears in plants [30]. Previous work [31] has shown that trees can send and act upon transmitted signals between them as a response to an insect attack. The communication can be performed within the same species or between different species [32]. Certain chemical compounds were identified as the main communication tool between the plants and between plants and insects [33]. Fungus has also been shown to create a web of communication between plants [34, 35].

Aerial signal transmission between the tree and plants has been explored [36, 37]. Furthermore, reversible memory in plants based on their surroundings was shown [38]. The memory is based on forecasting changes in the surroundings favorable to the plant and a complex changing control system [39, 40]. The knowledge structure which is useful to both the plant transmitting and the plant or animal receiving is still unknown [37].

Previous work showed the network between trees can be used for knowledge processing to implement decisions prioritizing the forest over a single tree regarding resource optimization. When there is a resection of a network of trees in a forest, a trail, each network part will try optimizing its overall access to light resources, represented by canopy tree coverage, independently. Following resection, forest activity showed behavior similar to neuron activity behavior [41].

This work presents the idea that the knowledge processing is not limited to a specific biological infrastructure, such as the brain, and similar patterns can also be observed in artificially generated network systems, which have their own knowledge processing power. If the purpose of each of these systems is different and the underlying processing structure is the same, then these knowledge processing systems might be much more common in Nature than previously thought.

## Invisible brain

"Invisible brain" suggests the idea that there are different domains where knowledge processing occurs [42]. Prior work identified that there is an additional layer of knowledge processing between the neuron network system and the brain [43, 44]. If there is another layer of knowledge processing and the knowledge processing occurs in multiple domains, then the issue of the processing platform becomes relevant. Is there a need for a brain or is the brain just one example of a processing system.

The main processing of the brain is based on spiking neurons sending messages communicating to other neurons. Assuming this process exists in other domains, it could be viewed in other knowledge-containing domains such as research publications. The assumption that research publication topics can represent a similar idea of neurons would require research topic publication spiking and sending communication signals to research topics. This process of communication of research topics published in articles can be monitored over multiple years. The process of signal processing in the brain can be compared to spiking interest in research topics across multiple areas.

What would happen if the way an Artificial Neural Network learns, which seems to show successful learning characteristics in tasks such as classifying handwritten digits or identifying images, is compared with the way Science learns new research directions and research topics over time. The work studied four knowledge processing patterns by accessing the structural patterns of natural and artificial networks responsible for such activities. This explorative study aims to test whether neuron networks responsible for knowledge processing in living organisms have any commonalities with manufactured networks with or without knowledge processing capabilities.

## Methods–knowledge processing comparison analysis

A set of comparison studies were performed to evaluate the existence of the theory of common knowledge processing patterns in networks of different systems. One study reviewed artificial neural networks. The second study reviewed the growing field of research knowledge. The third study checked what type of graph generating algorithms works well with the patterns identified. Finally, the fourth study reviewed known brain circuitries, connectomes, of various animals.

Several approaches have been studied to compare the knowledge processing patterns in both the biological and the artificial neural networks and show their similarities to the information reasoning and knowledge evolution in academic topic networks. The network is defined as a directional graph $G = \{V,E\}$ composed of vertices $V = \{v_1,v_2,...,v_n\}$, and edges $E = \{e_{i,j}|v_i, v_j \in V\}$. The mechanism of knowledge processing is studied using the four network measures shown below. The four measures evaluate the characteristics between nodes: topics, animal neural networks, artificial neural networks, and their relations represented by their connections or edges. The graph evaluation methods included:

- Average Degree = $2*|E|/|V|$.

  ○ The average degree of the graph is used to measure the number of edges compared to the number of nodes.

- Average PageRank = $\frac{1}{|V|} \cdot \sum_{v_i \in V} \sum_{v_j \in N^-(v_i)} \frac{PR(v_j)}{\deg^+(v_j)}$.

  ○ The average PageRank measures the average importance of the connected nodes in the network, where $|V|$ is the number of nodes, $N^-(v_i)$ is the inbound neighborhoods of the given node $v_j$, $\deg^+(v_j)$ is the outdegree of $v_j$, and $PR(v_j)$ is the PageRank of $v_j$.

- Average Clustering = $\frac{1}{|V|} \cdot \sum_{v_i \in V} 2 \cdot \frac{Triangle(v_i)}{\deg(v_i) \cdot (\deg(v_i)-1)}$

  ○ The average clustering coefficient measures the degree of node clusterability based on the number of triangles each node $v_i$ is a member of.

- Average Triangles = $\frac{1}{|V|} \cdot Triangle(V)$

○ The average number of triangles is measured to infer the approximated ratio of network completeness by measuring the ratio of smallest cliques in the network.

The evaluation measures provide domain-independent quality characteristic information for any graphs they are applied on. All the datasets used in the experiment share an identical graph format, therefore the measures can be used to objectively compare the characteristics of networks with vastly different origins. To achieve this, preprocessing has been performed on some of the artificial networks such as the Artificial Neural Networks which are usually fully connected to begin with. After the dataset preprocessing and graph generations, comparison between structures of different domains is reduced to a problem of comparison between different directed graphs.

## Patterns of decision making in artificial neural networks

Information processing and decision-making mechanisms are analyzed in a wide range of domains. First, the network patterns of the artificial neural networks are reviewed. The analysis checks which of the different characteristics of artificial neural networks repeat when the neural network evolves to achieve better classification results, or in other words, learns.

Three image classification datasets are used to differently train the artificial neural networks in order to analyze the common patterns observed during the decision-making process. MNIST (http://yann.lecun.com/exdb/mnist/) is a subset database of a larger NIST dataset, containing 60,000 training set examples of images of handwritten digits accompanied by 10,000 test set [45]. The 28 by 28 pixel images are centered for easier classification with 10 classes. Kuzushiji-MNIST (KMNIST) (http://codh.rois.ac.jp/kmnist/index.html.en) and Fashion-MNIST (FMNIST) (https://github.com/zalandoresearch/fashion-mnist) are respectively a Japanese Hiragana character classification dataset [46] and an apparel item classification dataset [47], each sharing the MNIST's format; 60,000 training set, 10,000 test set of 28x28 greyscale images with a total of 10 classes.

The Artificial Neural Network (ANN) is generated with one input layer with 28x28 = 784 neurons, followed by three hidden layers with 80, 40, and 20 neurons each and an output layer with 10 neurons. The Sigmoid function is used as hidden layers' activation functions where 0.5 is the median value, and the output layer neurons are set to have normalized categorical probabilities. The neural network is then trained using the training set with sparse categorical cross entropy as a loss function, accuracy as the metric function, and stochastic gradient descent Adam optimizer [48] as the optimization method.

Traditionally, fully connected layers are used; this is different from biological neural networks where the very structure of neuron-synapse connections represents knowledge processing patterns. The ANN has a total of 501,760,000 links between the neurons for given datasets, where many of them are obsolete in the decision making process. The patterns in ANN are therefore defined as the structures of *fired* neurons $V' = \{v_i | v_i \in V, output(v_i) > 0.5\}$, or neurons with the output value greater than the *activation_threshold = 0.5*, for each input.

The shared patterns shown during the decision-making process on different ANNs are analyzed. For each data set, the learning neural network characteristics were analyzed using the four graph evaluation measures; degree, PageRank, clustering, and triangles. The measures are averaged across each graph to generate a single value for any given graph. With multilabel classification datasets, the classification accuracy is recorded to show whether the found patterns are correlated to the correct decision-making process. Cohesion is also added as a relative quality measure for a set of subgraphs within the whole graph G
$(G' = \{V', E'\}$ where $E' = \{e_{i,j} | v_i, v_j \in V'\})$. It is defined as the relative quality of the subgraphs to the whole graph G or the ratio of edges internal to the subgraph

$|E'|/\{e_{i,j}|v_i \in V \cup v_j \in V, output_t(v_i|v_j) > 0.1\}$. The threshold variable $output_t(v_i|v_j) > 0.1$ is included in the formula to differentiate the otherwise complete graph; only nodes with output greater than the *graph_threshold = 0.1* are considered as the external neurons. Finally, the degree of knowledge processing maturity is captured by running with a different number of epochs during the ANN training to represent varying degree of training and network maturity. A total of 100 epochs are used for each dataset, and knowledge patterns at epochs up to 10, and every 10th up to 100 are recorded.

## Information processing mechanism on the academic topic networks

Next, the characteristics when information reasoning and decision making are performed in research are analyzed. This can be done by studying the change of entire fields of research over time and comparing the information processing mechanism to the processing mechanism of the neural network. In this case, the processing unit is a research topic in the field of research instead of a neuron.

Research activities share similarities with the knowledge processing in the brain; knowledge is propagated over the entities through their connections, which can result in the further propagation of modified knowledge. Similarities between the information processing mechanism on the research activity networks and the processing mechanism of the ANNs are reviewed, using topic networks as representations of research activities.

Topic network $T_i = \{V_i, E_i\}$ is defined as a graph with topics as vertices and topic co-occurrences as links observed by timeslot *i*. This is a cumulative network where the past research activities are stacked in $T_y$ to represent how the knowledge processing mechanism matured over time. The topics and topic co-occurrences are extracted from the Microsoft Academic Graph (MAG) dataset [49] for ease of access. It is one of the largest bibliographic data repositories with open access, and each publication within the dataset is linked to a built-in hierarchical topic ontology called Fields of Study, which is generated monthly by combining the preexisting knowledge base such as Wikipedia articles with state-of-the-art graph link analysis and convolutional neural networks [50]. The topics and topic co-occurrences are both present in the dataset, removing the necessity of further data preprocessing.

MAG contains more than 200 million publications, and analyzing the topic network as a whole is computationally too complex for this analysis. Subsets of the bibliographic dataset are used to build topic networks instead; publications under a specific journal are collected per the topic network. Journals are used as the publications in a shared set of common topics, while providing varying degrees of size, history, and research behaviors. *Nature* and *Science* are selected as large journals with broad research interest, while the *New England Journal of Medicine* (*NEJM)* and *Physical Review* (*Phys.Rev)* are selected as the top journals more focused on their respective research domains. *Cell* is selected to represent a top tier journal with a relatively short history. *Journal of High Energy Physics* (*HEP*), *Journal of Informetrics (JoI)*, and *Knowledge Based Systems (KBS)* are selected as topic-specific journals with the different research fields of particle physics, artificial intelligence, and informetrics, respectively.

The total of eight journals used covers a wide range of properties; the first year of the journal publication appearance ranges from *y = 1900* for *NEJM* to *y = 2007* for *JoI*. The four network measures for the knowledge processing mechanism are calculated for the ordered timeslot with index *i* for years *y ≤ 2020*, allowing an analysis of the mechanism maturity over time on one journal as well as the comparison between different journals at the same timeslots apart from their first appearances. The whole networks are considered; therefore the cohesion is not measured, as well as the classification accuracy.

The second set of comparisons reviewed for each set of selected journals the characteristics of the graph as the domain-specific topics mature over the years. The analyzed graph characteristics were the same except that there is only one value for each year for each journal. The comparison studies graph learning patterns over time as the number of topics in each journal domain increases. The increase in the number of edges and nodes was also evaluated.

## Similar patterns in randomly generated graphs

Next a study whether random graph generating algorithms used to create networks share similar characteristics to the artificial neural network or the journal topic domain knowledge was performed. Similar characteristics were reviewed to find whether one of the graph generating methods could serve as a predictive tool for any of these knowledge processing patterns.

Information processing is controlled by the basic mechanism of knowledge processing. The four measures are found on randomly generated graphs to identify the shared basic patterns in not only the ANNs, the topic networks, but also any network structures.

Four graph generating algorithms are reviewed. The Barabási–Albert (BA) model is an algorithm for generating a random scale-free network $G_{n,e}$ using the BA preferential attachment mechanism [51] with $n$ nodes each connected to $e$ existing nodes. $V = \{v_1, v_2, \ldots, v_n\}$, $v_{e+1}$ is first linked to $\{v_1, v_2, \ldots, v_e\}$ to initialize the graph. Node set $\{v_{e+2}, v_{e+3}, \ldots, v_n\}$ is then iteratively added to the graph each with $e$ edge to the existing node $v_i$ each with probability $p_{v_i} = \deg(v_i)/\sum_n \deg(v_n)$. Two variants of the Erdős-Rényi model [52], $G_{n,p}$ and $G_{n,m}$, are used. $G_{n,p}$ is generated by randomly generating links between $n$ vertices $V = \{v_1, v_2, \ldots, v_n\}$, where each possible link $(v_i, v_j)$ has an independent probability $p$ of being created. $G_{n,m}$ is uniformly selected from a set of all non-isomorphic graphs with $n$ nodes and $m$ edges. Last, random regular graph $G_{n,d}$ is generated as a $d$-regular graph with $n$ vertices which denotes the probability space of all $d$-regular graphs on $n$ vertices, where $3 \leq d < n$ and $n \cdot d$ is even [53]. The graph generating algorithms are run 20 times to retrieve average values of the four measures, with $n = 200$, $p = 0.015$, and $e = d = 20$.

## Knowledge processing patterns in animal neuron-synapse structures

From there, the most common acceptable process of thinking attributed to the biological neuron is evaluated. The work looks at animal neuron systems, which in the literature were explored and their full neural network mapped. Both the artificial and the biological networks rely on a similar basic mechanism, the neuron. The analysis of the knowledge processing characteristics found both in artificial neurons and in research fields shows that the systems are able to process more information when as the systems get more complex. When compared to multiple animal brain system structures at different levels of brain complexity, these systems show similarities.

Neurons are viewed as the basic cells that process and transmit information, and this study is conducted to compare the similarities between the information processing mechanism in neuron-based systems and non-neuron-based systems.

The last study reviews the network configuration in animals. This includes the Caenorhabditis elegans (roundworm) and Fruit Fly, for which the number of neurons and the brain synapses have been fully mapped, as well as eight animals with estimated number of neurons whose connections are projected with random graph generation algorithms using a various range of neuron-synapse ratios. The characteristics of the fully mapped and algorithmically generated animal neural networks are compared to the naturally occurring animal networks.

There are a limited number of openly available connectomes; two connectomes, or neural maps, are used in the study. A neural connectivity of Caenorhabditis elegans (roundworm)

nerve system by Wormatlas (https://www.wormatlas.org/neuronalwiring.html) contains 6,417 connections between 281 neurons [54]. The neuron types are separated into single/poly-synaptic connections, electric junctions, and neuromuscular junctions, when each can be interpreted as the hidden layers, input layer, and output layer in ANN structures. Sending and receiving connections refer to duplicate synapses; therefore the Receiving and Receiving-Poly types are removed from the dataset, resulting in a total of 2,405 non-overlapping connections. The hemibrain connectome [55] is the largest reconstructed synaptic connectome dataset to date, storing the neuron connectivity data for the central part of a fruit fly brain for tasks including associative learning, flight navigation, and sensory input processing. The dataset contains around 25,000 neurons, and of those 21,733 neurons were extracted with 2,872,500 synaptic connections.

The number of connections between neurons and types of neurons is ignored to build single-layered, unweighted graphs following the same format from the previous comparisons. The differences in the structures of actual neural maps and randomly generated graphs are studied by generating two sets of random networks sharing the same graph properties of two actual neural maps. $G_{n,p}$ of the Erdős-Rényi model is used where the edge probability $p$ is calculated as $p = 2 \cdot |E|/(|V|*(|V|-1))$, resulting in $p = 0.0611$ for the roundworm and $p = 0.0122$ for the fruit fly.

Comparing the measurement of two neural maps is limited in that there are no mid-points from which to infer the gradual changes. Intermediate graphs are generated by applying the $G_{n,p}$ Erdős-Rényi model with $n$ and $p$ values around that of the roundworm and the fruit fly. Eight animals with an estimated number of neurons are selected, with $n$ ranging from 200 to 18,000 as shown in Table 1. The differences between $p$ for two known neural maps are not similar; therefore ten $p$ values between the two $p$s are used to generate ten graphs for each animal.

Next, the previously mapped Caenorhabditis elegans (roundworm) and Fruit Fly were used to build a network (using the mapped nodes/links). The Erdős-Rényi Gn,p algorithm was implemented to generate a graph based on a similar number of neurons and synapses using the same number of n and p. Then, the algorithm was tested on additional different animals to generate a graph based on a similar number of neurons/synapses.

Two graphs were generated using n and p for each insect, where
n = the neuron size

- n(worm) = 281

- n(fruitfly) = 21,733

  p = number of edges / possible edges (n*(n-1)/2)

- p(worm) = 0.0611

- p(fruitfly) = 0.0122

Table 2 shows the degree, PageRank, clustering, and triangles of the worm and fruit fly compared to the graph generated by the Erdős-Rényi algorithm. The average degree and

**Table 1. List of animals used as intermediates between roundworm and fruit fly with estimated number of neurons.**

| Animal | # Neurons | Animal | # Neurons |
|---|---|---|---|
| Asplanchna brightwellii (rotifer) [56] | 200 | Megaphragma mymaripenne [60] | 7,400 |
| Ciona intestinalis larva (sea squirt) [57, 58] | 231 | Medicinal leech [61] | 10,000 |
| Caenorhabditis elegans (roundworm) [54] | 302 | Pond snail [62] | 11,000 |
| Jellyfish [59] | 5,600 | Sea slug [63] | 18,000 |

**Table 2. Worm and fruit fly versus generated graph.**

|  | #neurons | #synapses | avg_degree | avg_pagerank | avg_clustering | avg_triangles |
|---|---|---|---|---|---|---|
| Worm | 281 | 2,405 | 17.1174 | 0.0036 | 0.3460 | 47.5943 |
| Fruitfly | 21,733 | 2,872,500 | 264.3445 | 4.60E-05 | 0.3137 | 12605.9781 |
| Generated_worm | 281 | 2,350 | 16.7260 | 0.0036 | 0.0611 | 8.5836 |
| Generated_fruitfly | 21,733 | 2,873,553 | 264.4414 | 4.60E-05 | 0.0122 | 425.5065 |

PageRank are directly dependent on the number of neurons and the synapses, and therefore they are consistent with both the natural and artificially generated graph. The interesting values are the clustering, which seems to decrease as the animal processing ability, defined by the graph as the number of neurons and synapses, increases. Conversely, the number of triangles increases as the animal is considered to have a more complex knowledge processing mechanism. In both cases, the Erdős-Rényi algorithm shows similar patterns although the scale is not identical for the clustering and triangles.

The changes in the four properties basically come from a larger size and increasing complexity so it can be stated that the changes in the properties of generated graphs would follow those of the real neural maps. The comparison shows that the changes with larger n are consistent with the neuron maps. As long as the p is known, the same algorithm can be used with different numbers of neurons and similar results can be expected.

Based on the previous data, the graph properties were projected for an additional eight animals for which the number of neurons was previously published (Table 1). The animals were selected due to their number of neurons, which is in the range of the worm and the fruit fly for which all the data is available. The projected structural properties use a randomly generated graph with varying p values. Ten p values are uniformly chosen between p of fruit fly (0.012164) and p of worm (0.061134).

## Results

The generated graphs are analyzed to detect unique and shared characteristics between biological and artificial knowledge processing networks, with the random networks as a baseline with the simplistic nature and the connectomes as the biological structure for the knowledge processing functions. Each graph is first analyzed independently with the evaluation measures, and the commonalities and differences in their measures are then reviewed. Artificial networks with similar evaluation measures to biological networks are interpreted as networks sharing knowledge processing mechanisms to the neural and brain networks.

Fig 1 shows the graph learning characteristics as the artificial neural network learns. The X-axis represents the growing number of epochs, and the Y-axis shows the accuracy (a) The results show that the accuracy increases and stabilizes for all datasets; the correlation coefficient of 0.4322 between the accuracy of the epoch count of up to 10 shows medium correlation during the initial stage of network maturation. In (b) the results show that the average degree decreases and stabilizes. PageRank (c) also shows a change, which is an increase, but the increase itself is small due to the characteristics of PageRank. The smaller changes are also reflected in its jagged patterns, which show nearly no correlation (-0.0566) and statistically significant mean value differences (T-test p value = 0) with the accuracy. Similarly, all other measures showed low to no correlation to the accuracy with the clustering showing the highest value (0.2533). This indicates that the maturity of the knowledge processing network is not governed by a single graph structure measure. The cohesion (d) and clustering (e) do not present consistent behavior as the number of epochs increases and therefore are less representative

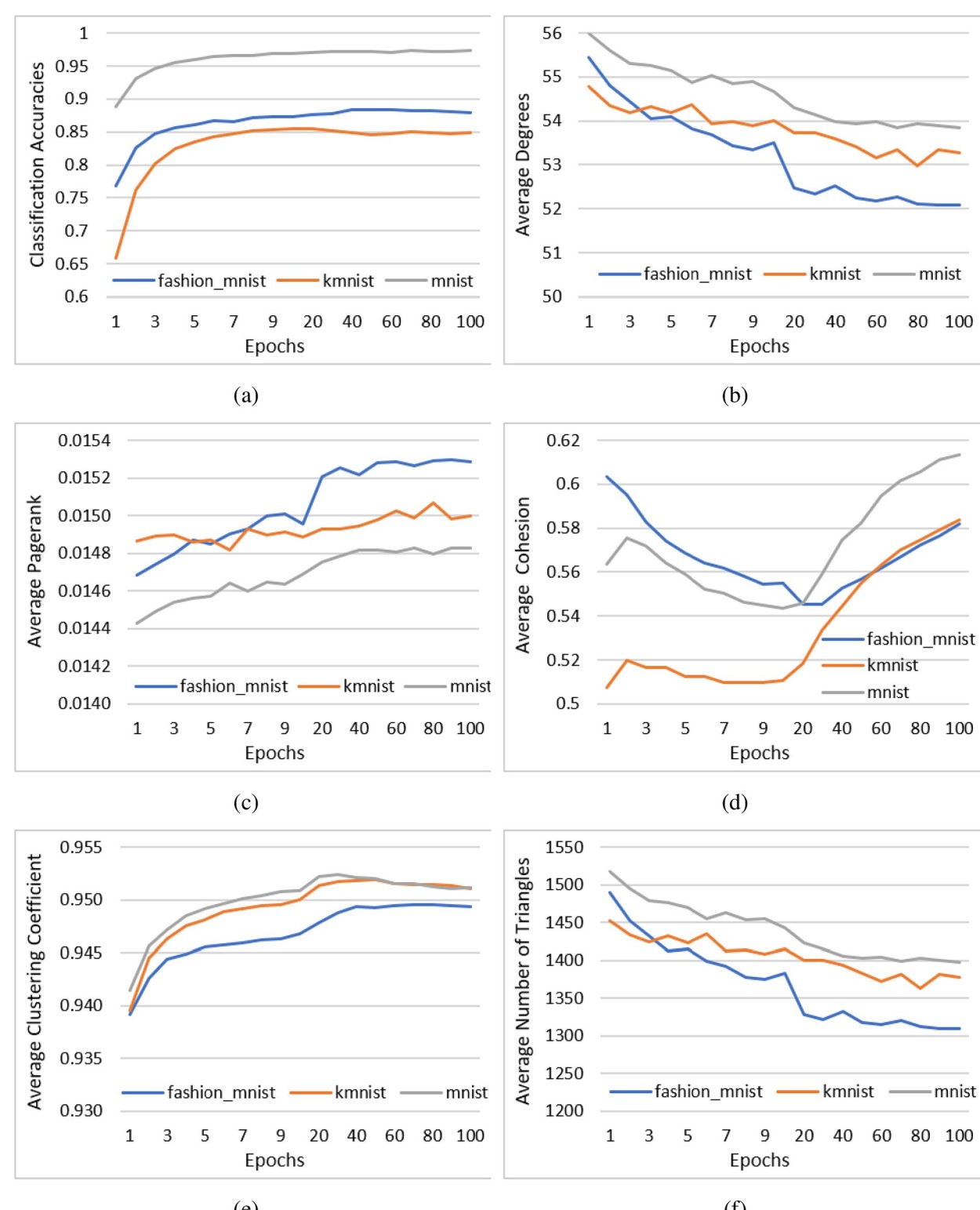

**Fig 1. Graph learning characteristics as the artificial neural network learns as shown by the average properties of neural networks.**

measurements of the change of graph characteristics. Finally, the number of triangles (f) displays the most consistent decrease and stabilization of the number of triangles along with the degree (b) over epochs with near identical patterns (correlation value of 0.9963, with statistically significant mean normalized differences with p = 0). This is a natural behavior of growing networks as the change in the average degrees directly affects the average number of triangles each node can be a member of.

Fig 2 describes the research activity as an "invisible brain" according to topic behavior and reviews graph learning characteristics in the past 120 years of advancement, with 1900 to 2020 as the X-axis. (a) and (b) show the increase in the number of nodes and edges accordingly. Most publication outlets present almost linear growth in nodes and then exponential growth in edges to the more historic publications with extremely high correlation (0.9755). The only exception is Physical Review which seems to have artificially limited the topics, and as a result the nodes of publication have been limited to a fixed number in 1970 which influences all the results. Average degree (c) shows consistent growth, while PageRank (d) shows a very fast decrease and stabilization shortly after each new journal is created. The clustering (e) consistently has linear decline as the years advance. The number of triangles (f) presents a linear increase similar to the degree but represents the most consistent characteristic, which describes the increase in knowledge as the number of topics represented grows in each of the publications. While the degree and the triangles show the same high correlation (0.9267) as they did in the previous example, they show an inverse pattern over time with increasing values as opposed to in Fig 1. This represents a different knowledge processing pattern; while the classification ANNs matured by disseminating their functions over network structures, journal publications instead matured by centralizing on key topics. Such differences can be seen between the relationship between the four evaluation measures and their sizes as well; while ANOVA shows statistically significant differences (p = 0) between all six variables, their evolution patterns show different similarities with diminishing PageRank and clustering values over the years.

Fig 3 compares the four different graph generating algorithms, Barabási–Albert $Gn,e$ and Erdős-Rényi $G_{n,p}$, $G_{n,m}$, and random regular graph $G_{n,d}$, to evaluate which method has the most similar characteristics to the artificial neural network and the "invisible brain" represented by the growing network of research topics. The randomly generated graphs were expanded over 20 iterations, adding 200 nodes at each iteration, resulting in the total of 4,000 nodes in the final graph. The results show that the average degree (a) is constant in $G_{n,m}$ and $G_{n,d}$ and almost constant except for the beginning in the Barabási–Albert with the iteration count as the X-axis. However, as the graph iterations increase all the way to twenty, the Erdős-Rényi $G_{n,p}$ method presents a consistent linear increase in the average number of triangles (c). This is because only $G_{n,p}$ has an edge generation variable dependent on the population sizes. The BA model dictates the degree with the existing node connection variable e, which results in edge overlaps in lower iterations. $G_{n,m}$ and $G_{n,d}$ each dictate their edge generations with a fixed edge size e and regular graph with neighbor count d, resulting in a static degree during graph expansion. Only $G_{n,p}$ results in a linear growth in the degree as the population of possible edges increases exponentially. The number of edges therefore increases exponentially with static edge probability p resulting in linear increase in node degrees.

Similarly, for the average clustering all three methods except for Erdős-Rényi $G_{n,p}$ present a logarithmic decrease in value, while Erdős-Rényi $G_{n,p}$ presents just the opposite, with increasing growth in the average number of triangles in the graph. Comparing the PageRank values of each of the four methods (b) shows the exact same results. The PageRank is a characteristic used to generate the graphs in all four methods and therefore cannot be used as a differentiating feature. Finally, the graph average clustering (d) displays similar behavior to the average

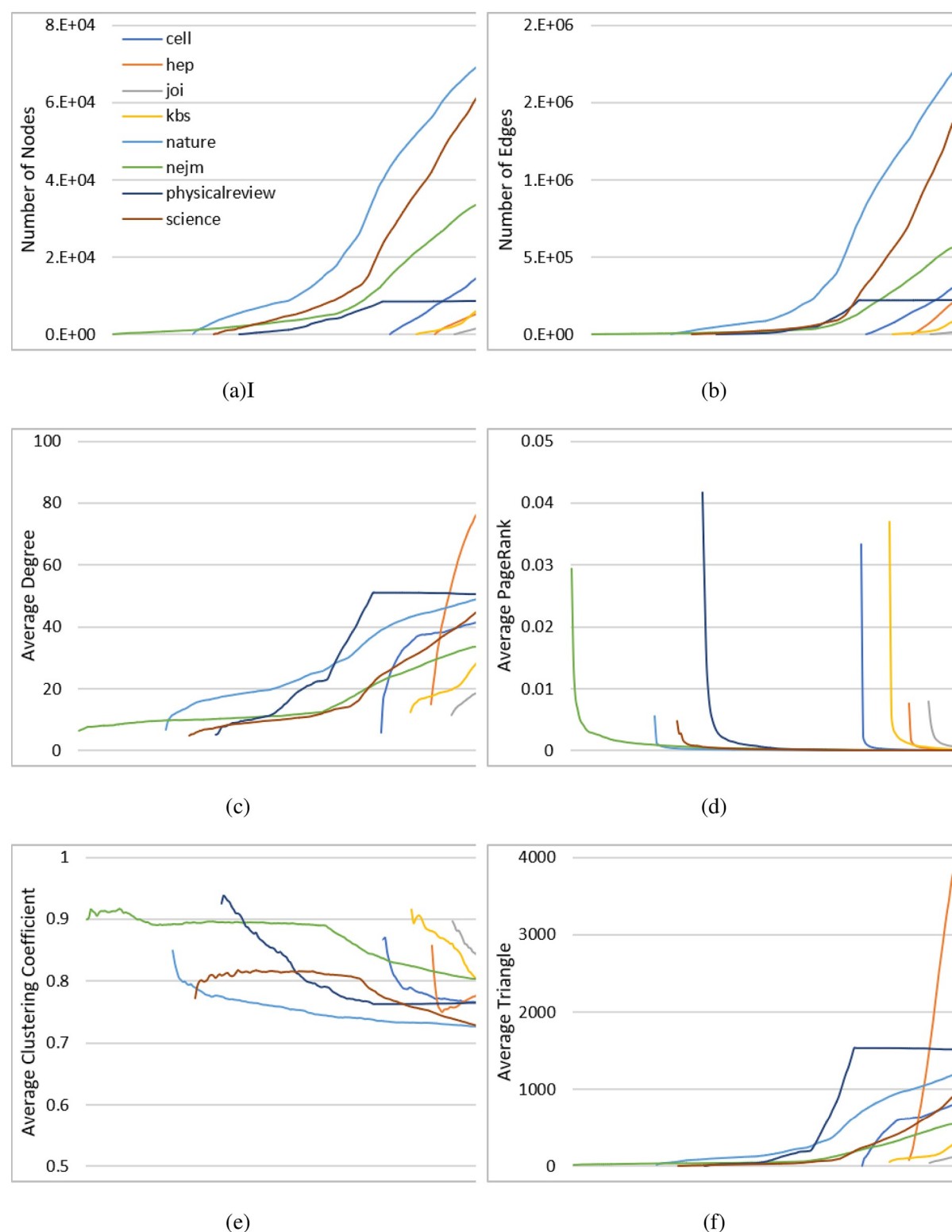

**Fig 2. Research activity ("invisible brain") graph learning characteristics.**

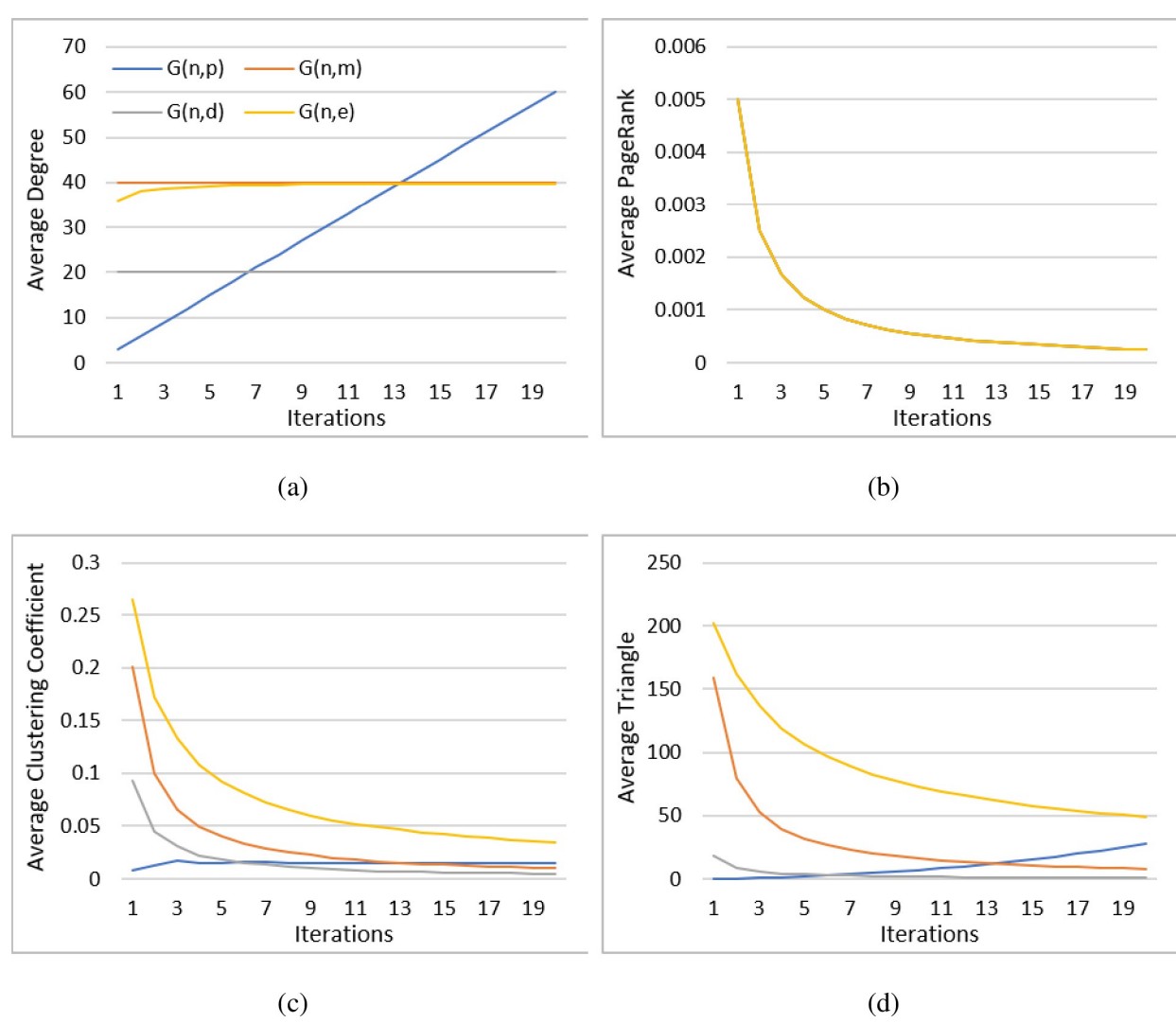

**Fig 3. Graph generated characteristics for multiple algorithms.**

number of triangles of all three methods having a fast logarithmic decrease in value except for the Erdős-Rényi $G_{n,p}$, which presents an expanding increase throughout all twenty iterations, although the increase slows down after the third iteration.

Fig 4 presents the projected graph properties for the different animals. Ten neural maps were artificially generated for each animal in Table 1 using ten uniformly distributed edge completion ratios ranging from p(fruitfly) = 0.012164 to p(worm) = 0.061134. With their estimated number of neurons as the number of nodes, edges between the neurons are generated using the ratio values (p1, p2, . . ., p10) to reflect the varying degree of connectome complexity. The two values shown as the black cross represent the values of the true neural maps of fruit fly and the worm. Eight animals from Table 1 are represented by their estimated number of neurons on the X-axis. All Y-axis values are presented in logarithmic scale. All the iterations showed that larger p value and larger n value both result in higher synapses (a), higher average degree (b), and higher average triangles (d). Average clustering (c) fluctuated more by the p value and was less affected by the n value. As the number and distribution of edges are governed by a graph generation algorithm, all three evaluation measures used in the analysis

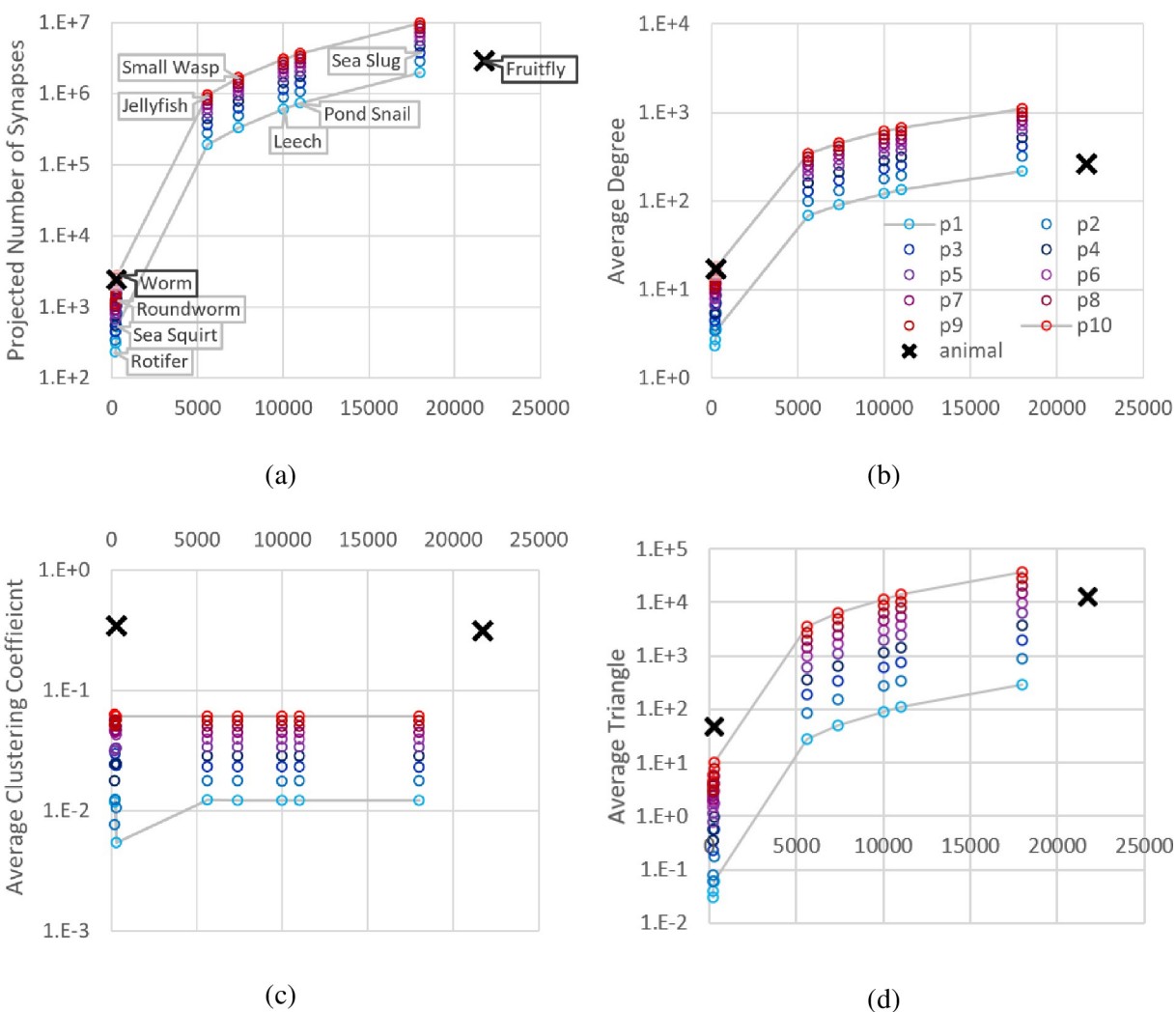

**Fig 4. Projected graph properties for different animals, with the number of neurons as the X-axis.**

showed statistically insignificant differences with ANOVA p-value of 1 across the eight animals, indicating that the main factor is the number of neurons.

## Discussion—knowledge regardless of the brain

Knowledge processing appears to be relatively common in many areas, and neurons, whether created naturally or artificially, do not seem to be required as the basic mechanism for enabling the information processing. This leads to the question of what are the most basic rules, or graph characteristics, required to perform information pattern processing. The results present

**Table 3. Knowledge networks comparison summary.**

|  | avg_degree | avg_pagerank | avg_clustering | avg_triangles |
|---|---|---|---|---|
| Artificial neural network classification with increasing epochs | Down | Up | Up | Down |
| **Topic networks with incrementing years** | **Up** | **Down** | **Down** | **Up** |
| Erdős-Rényi Random networks with increasing size | Up | Down | Down | Up |
| **Animal neuron maps with increasing #neurons** | **Up** | **Down** | **Down** | **Up** |

these basic sets of rules required to process information and show these sets of rules using graph analysis techniques to view how well they perform under different conditions.

Finally, a comparison of the characteristics between all previous studies is presented, reviewing only the changes in the trend as the **knowledge network** gets larger. The summary is displayed in Table 3. The artificial neural network seems to be consistent with most of the graph generation algorithm trends regarding PageRank, clustering, and triangles but not degree. However, if the Erdős-Rényi graph generating algorithm is used, then similar characteristics to the topic network, graph generating algorithm, and animal neuron network are shown. This is somewhat counterintuitive, since while usually artificial neural networks present good results in finding patterns in knowledge, in this case the characteristics of the artificial neural networks present opposite trends of the graph pattern change, compared to the other domains. The opposite trends could be explained by the fact that the artificial neural networks used today are generated using most of the common graph generation algorithms. However, graph generating algorithms such as Erdős-Rényi present trends which are more consistent with both the increasing topic network and the natural configurations of the specimens reviewed. An alternative configuration of artificial neural networks might yield trends that are more consistent with natural networks.

The question of how knowledge is represented and whether it has a basic structure is analyzed. The analysis was performed on graphs of artificial neural networks, research topic expanding graphs, artificially generated graphs, and real neural network structures in animals. There appears to be a method to the madness, and specific graph properties increase in a predefined structure that can be represented by graph generating algorithms.

The overall result analysis shows an increase in the average degree, which measures the number of edges compared to the number of nodes, and an increase in the average number of triangles, measuring the ratio of smallest cliques in the network. This shows that as more knowledge is represented by the network the network becomes more tightly knitted.

At the same time there is a decrease in the PageRank, which measures the average importance of the connected nodes in the network, and a decrease in the average clustering, which measures node clusterability based on the number of triangles each node is a member of. These results show that specific nodes become less unique or important as the knowledge in the network increases and the nodes representing specific knowledge are more widely shared or less clustered together as the network increases.

## Conclusion and further research

Basic patterns of knowledge processing are found across systems which appear to be essentially different and each one serves a different purpose. Furthermore, some of these systems are naturally created biologically based systems, while others are artificial systems created by humans and work on electronic circuits. The similarity in patterns of knowledge processing systems raises the question of how unique are humans in processing knowledge.

Further research can improve the similarity in values of properties such as clustering by building more tailored algorithms for representing the graph growth as the knowledge represented increases. Other artificial neural networks might fit the patterns found in topic networks, random graph generation methods, and mapping animals with increasing number of neurons. Analyzing convolution neural networks, deep neural networks, and graph neural networks for these characteristics might yield interesting results. In addition, as additional research works identifying the sizes of different animal neural networks increase, verifying the different animal neural network sizes, the properties projecting the graph characteristics can be adjusted.

## Author Contributions

**Conceptualization:** Aviv Segev.

**Data curation:** Sukhwan Jung.

**Formal analysis:** Aviv Segev, Sukhwan Jung.

**Investigation:** Aviv Segev, Sukhwan Jung.

**Methodology:** Aviv Segev, Sukhwan Jung.

**Software:** Sukhwan Jung.

**Validation:** Aviv Segev, Sukhwan Jung.

**Visualization:** Sukhwan Jung.

**Writing – original draft:** Aviv Segev, Sukhwan Jung.

**Writing – review & editing:** Aviv Segev, Sukhwan Jung.

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
