## [Decision Letter · Decision Letter 0]

20 Jun 2023

PONE-D-23-15020

Common Knowledge Processing Patterns in Networks of Different Systems

PLOS ONE

Dear Dr. Segev,

Thank you for submitting your manuscript to PLOS ONE. After careful consideration, we feel that it has merit but does not fully meet PLOS ONE’s publication criteria as it currently stands. Therefore, we invite you to submit a revised version of the manuscript that addresses the points raised during the review process.

We look forward to receiving your revised manuscript.

Kind regards,

Nebojsa Bacanin

Academic Editor

PLOS ONE

Journal Requirements:

Additional Editor Comments:

Dear Authors,

according to reviewers' assessment and my own evaluation, I advise giving this manuscript a chance for improvements. Therefore, my decision is major revision.

Warmest,

Nebojsa Bacanin

 Dear Authors, please revise your manuscript carefully according to your reviewers' comments. Please provide concise response letter replying to each issue raised by reviewers'. 

I would like to add that more rigid statistical validation is required in order to prove robustness of your approach. Please choose carefully adequate parametric or non-parametric statistical test.

Thank you.Warmest regards,AE

Additional Comments from the Editorial Staff:

We note that one reviewer has recommended that you cite specific previously published works. As always, we recommend that you please review and evaluate the requested works to determine whether they are relevant and should be cited. It is not a requirement to cite these works. We appreciate your attention to this request.

Reviewers' comments:

Reviewer's Responses to Questions

**Comments to the Author**

1. Is the manuscript technically sound, and do the data support the conclusions?

Reviewer #1: Partly

Reviewer #2: Yes

2. Has the statistical analysis been performed appropriately and rigorously? 

Reviewer #1: N/A

Reviewer #2: N/A

3. Have the authors made all data underlying the findings in their manuscript fully available?

Reviewer #1: Yes

Reviewer #2: No

4. Is the manuscript presented in an intelligible fashion and written in standard English?

Reviewer #1: Yes

Reviewer #2: Yes

5. Review Comments to the Author

Reviewer #1: 1. Avoid using we/our throughout the paper.

2. Third sentence in the Introduction: Can there be... is not clear, please rephrase it.

3. Fourth paragraph in the Introduction, starting with What if we compare... if those are questions, please add question marks (?)

4. State the research question, and list the main contributions clearly in the Introduction.

5. Provide paper structure at the end of introduction.

6. The literature survey should include the swarm intelligence/machine learning section, as it is considered to be the state of the art. You can include the following:

https://www.mdpi.com/2075-1680/12/3/266

https://onlinelibrary.wiley.com/doi/abs/10.1002/cpe.6629

https://www.sciencedirect.com/science/article/pii/S2210670720308842

https://dl.acm.org/doi/abs/10.1145/3546194

https://link.springer.com/chapter/10.1007/978-981-16-5348-3_54

7. The experiments should be elaborated in more details.

8. The purpose of the experiments should be highlighted.

9. The discussion is very limited, it should be more elaborate.

10. The conclusions should be clear and backed up with experimental results.

Reviewer #2: This is a remarkable study aiming to identify the underlying structures in knowledge processing across different domains, specifically in biological neuron activity and artificial neural networks. Your exploration of common data processing patterns in biological systems and human-made knowledge-based systems is thought-provoking and has the potential to redefine our understanding of the inherent nature of knowledge processing.

Strengths:

• Cross-Domain Analysis: Your study interestingly examines the commonalities in knowledge processing across a broad range of systems, from artificial neural networks to animal connectomes. This cross-disciplinary approach provides a unique perspective and adds a lot of value to the research.

• Dynamic Pattern Exploration: The exploration of changes in system patterns over time is a crucial contribution to the understanding of the evolution of knowledge processing systems.

• Groundbreaking Research Question: The question raised, about how unique knowledge processing is in humans if similar patterns are found in both natural and artificial systems, is compelling and invites further research.

Suggestions for Improvement:

• More in-depth analysis of related works. Add a separate Related Works section to discuss as currently it is missing. Here are some suggestions to start with: Topic classification of online news articles using optimized machine learning models. Computers, 12(1); Relation-aware weighted embedding for heterogeneous graphs. Information Technology and Control, 52(1), 199-214. Large scale community detection using a small world model. Applied Sciences, 7(11).

• Clearer Explanation of Methods: While your methodology is intriguing, a more detailed explanation of the procedures used, such as the graph analysis techniques and the criteria for comparing different systems, would enhance the understanding and replicability of your research.

• Discussion on Degree Consistency: Your observation that artificial neural networks align with most trends in graph generation algorithms regarding PageRank, clustering, and triangles but not degree is interesting. A more in-depth discussion about the implications of this inconsistency would be useful.

• Future Directions: In the conclusion, you mention potential areas for further research. Fleshing out these ideas a little more would provide an excellent direction for future work in this field.

In conclusion, this research study provides fascinating insights into the commonalities in knowledge processing across different systems. The findings from your study have the potential to make a significant impact on the field, and I am looking forward to seeing how this research evolves in the future.

6. PLOS authors have the option to publish the peer review history of their article (what does this mean?). If published, this will include your full peer review and any attached files.

Reviewer #1: No

Reviewer #2: No

---

## [Decision Letter · Decision Letter 1]

4 Aug 2023

Common Knowledge Processing Patterns in Networks of Different Systems

PONE-D-23-15020R1

Dear Dr. Segev,

We’re pleased to inform you that your manuscript has been judged scientifically suitable for publication and will be formally accepted for publication once it meets all outstanding technical requirements.

Kind regards,

Nebojsa Bacanin

Academic Editor

PLOS ONE

Additional Editor Comments (optional):

Dear Authors,

thank you for revising your manuscript.

All the best,

AE

Reviewers' comments:

Reviewer's Responses to Questions

**Comments to the Author**

1. If the authors have adequately addressed your comments raised in a previous round of review and you feel that this manuscript is now acceptable for publication, you may indicate that here to bypass the “Comments to the Author” section, enter your conflict of interest statement in the “Confidential to Editor” section, and submit your "Accept" recommendation.

Reviewer #1: All comments have been addressed

Reviewer #3: All comments have been addressed

2. Is the manuscript technically sound, and do the data support the conclusions?

Reviewer #1: (No Response)

Reviewer #3: Yes

3. Has the statistical analysis been performed appropriately and rigorously? 

Reviewer #1: (No Response)

Reviewer #3: Yes

4. Have the authors made all data underlying the findings in their manuscript fully available?

Reviewer #1: (No Response)

Reviewer #3: Yes

5. Is the manuscript presented in an intelligible fashion and written in standard English?

Reviewer #1: (No Response)

Reviewer #3: Yes

6. Review Comments to the Author

Reviewer #1: (No Response)

Reviewer #3: The manuscript was well revised. The quality has been improved. I recommend to accept this manuscript for publication.

7. PLOS authors have the option to publish the peer review history of their article (what does this mean?). If published, this will include your full peer review and any attached files.

Reviewer #1: No

Reviewer #3: No

---

## [Editor Report · Acceptance letter]

25 Sep 2023

PONE-D-23-15020R1 

Common Knowledge Processing Patterns in Networks of Different Systems 

Dear Dr. Segev:

I'm pleased to inform you that your manuscript has been deemed suitable for publication in PLOS ONE. Congratulations! Your manuscript is now with our production department. 

Kind regards, 

on behalf of

Dr. Nebojsa Bacanin 

Academic Editor

PLOS ONE